# Contrasting Epidemiology and Population Genetics of COVID-19 Infections Defined by Multilocus Genotypes in SARS-CoV-2 Genomes Sampled Globally

**DOI:** 10.3390/v14071434

**Published:** 2022-06-29

**Authors:** Felicia Hui Min Chan, Ricardo Ataide, Jack S. Richards, Charles A. Narh

**Affiliations:** 1Central Clinical School, Monash University, Melbourne, VIC 3004, Australia; f.chan@nus.edu.sg; 2Burnet Institute for Medical Research, Melbourne, VIC 3004, Australia; jack.r@zipdiag.com; 3Department of Infectious Diseases, University of Melbourne at the Peter Doherty Institute for Infection and Immunity, Melbourne, VIC 3004, Australia; ataide.r@wehi.edu.au; 4Population Health and Immunity Division, Walter and Eliza Hall Institute, Melbourne, VIC 3052, Australia; 5Department of Medicine, University of Melbourne, Victoria, VIC 3052, Australia; 6Department of Infectious Diseases, Monash University, Melbourne, VIC 3004, Australia; 7ZiP Diagnostics Pty Ltd., Collingwood, Melbourne, VIC 3066, Australia

**Keywords:** COVID-19, SARS-CoV-2, epidemiology, genetics, multilocus, evolution, linkage, mutation and transmission

## Abstract

Since its emergence in 2019, SARS-CoV-2 has spread and evolved globally, with newly emerged variants of concern (VOCs) accounting for more than 500 million COVID-19 cases and 6 million deaths. Continuous surveillance utilizing simple genetic tools is needed to measure the viral epidemiological diversity, risk of infection, and distribution among different demographics in different geographical regions. To help address this need, we developed a proof-of-concept multilocus genotyping tool and demonstrated its utility to monitor viral populations sampled in 2020 and 2021 across six continents. We sampled globally 22,164 SARS-CoV-2 genomes from GISAID (inclusion criteria: available clinical and demographic data). They comprised two study populations, “2020 genomes” (*N* = 5959) sampled from December 2019 to September 2020 and “2021 genomes” (*N* = 16,205) sampled from 15 January to 15 March 2021. All genomes were aligned to the SARS-CoV-2 reference genome and amino acid polymorphisms were called with quality filtering. Thereafter, 74 codons (loci) in 14 genes including *orf1ab* polygene (*N* = 9), *orf3a*, *orf8*, nucleocapsid (N), matrix (M), and spike (S) met the 0.01 minimum allele frequency criteria and were selected to construct multilocus genotypes (MLGs) for the genomes. At these loci, 137 mutant/variant amino acids (alleles) were detected with eight VOC-defining variant alleles, including N KR_203&204_, *orf1ab* (I_265_, F_3606_, and L_4715_), *orf3a* H_57_, *orf8* S_84_, and S G_614_, being predominant globally with > 35% prevalence. Their persistence and selection were associated with peaks in the viral transmission and COVID-19 incidence between 2020 and 2021. Epidemiologically, older patients (≥20 years) compared to younger patients (<20 years) had a higher risk of being infected with these variants, but this association was dependent on the continent of origin. In the global population, the discriminant analysis of principal components (DAPC) showed contrasting patterns of genetic clustering with three (Africa, Asia, and North America) and two (North and South America) continental clusters being observed for the 2020 and 2021 global populations, respectively. Within each continent, the MLG repertoires (range 40–199) sampled in 2020 and 2021 were genetically differentiated, with ≤4 MLGs per repertoire accounting for the majority of genomes sampled. These data suggested that the majority of SARS-CoV-2 infections in 2020 and 2021 were caused by genetically distinct variants that likely adapted to local populations. Indeed, four GISAID clade-defined VOCs - GRY (Alpha), GH (Beta), GR (Gamma), and G/GK (Delta variant) were differentiated by their MLG signatures, demonstrating the versatility of the MLG tool for variant identification. Results from this proof-of-concept multilocus genotyping demonstrates its utility for SARS-CoV-2 genomic surveillance and for monitoring its spatiotemporal epidemiology and evolution, particularly in response to control interventions including COVID-19 vaccines and chemotherapies.

## 1. Introduction

SARS-CoV-2, the causative virus of the COVID-19 pandemic, emerged in December 2019 and has since infected more than 500 million and killed over 6 million people globally. Public health interventions including testing and isolation of infected individuals, wearing of face masks, and the rollout of vaccines in 2021 in many affected countries have contributed to significant declines in the transmission and incidence of SARS-CoV-2 infections [1,2]. Surveillance data, including phylogenomic analysis of clinical infections, has shown that the virus has evolved with highly transmissible variants, i.e., variants of concern (VOCs), including Alpha, Beta, Gamma, Delta, and Omicron. These variants have driven successive waves of transmission in different continents and territories [3,4,5]. Other reports have associated variant-specific outbreaks with different demographics and clinical forms of COVID-19, but whether these factors can explain the case disparities reported globally has not been thoroughly investigated [6,7].

Phylogenomic analysis of SARS-CoV-2 whole genomes has been used to differentiate variants when source-tracking infections and inform public health control efforts. However, the former analysis can be computationally challenging when dealing with thousands of genomes [8]. Additionally, most single-nucleotide polymorphisms (SNPs) in the genome are evolutionarily neutral and, thus, of little virological relevance [9]. Genome-wise, two-thirds of SARS-CoV-2′s ~30 kb positive-stranded RNA constitute the *orf1ab* polygene, which encodes 16 non-structural proteins (nsp1–16) associated with viral replication. The remaining one-third encodes structural proteins such as the spike glycoprotein (S), matrix (M), envelope (E), and nucleocapsid (N), and accessory proteins including ORF3a and ORF8 [10].

VOCs such as the Indian Delta variant have been associated with the presence of “evolutionary-relevant” or “adaptive” mutations, including L452R, T478K, N501Y, and D614G in the S protein, a target of cell- and vaccine-mediated immunity [4,11]. In the United Kingdom, the B.1.1.7 lineage (Alpha variant) with the spike mutations of E484K, D614G, and N501Y was shown to spread more rapidly among children and young people (<20 years) than in adults [12]. In addition, several non-S mutations including *orf1ab* P4715L, *orf3a* G251V, *orf8* L84S, N R203K, and N G204R have been implicated in high infectivity and replication [13,14,15] and have been designated for clade classifications of VOCs including 20I (Alpha), 21A (Delta), 21B (Kappa), and 20H (Beta) [16]. The epidemiological diversity and frequency of these mutations have been shown to vary spatiotemporally among different geographical regions [17], making the loci carrying these mutations “informative” for monitoring the viral transmission dynamics and evolution in different populations [10,15,18,19]. Genetic tools and approaches are needed to identify other loci that can be used to define and monitor SARS-CoV-2 populations.

Multilocus genotyping of amino acid polymorphisms at putatively adaptive loci in SARS-CoV-2 could be a useful genetic tool to monitor the viral epidemiology, transmission dynamics, and evolution in different populations and regions worldwide. To reduce the complexity in the SARS-CoV-2 genomic space, multilocus genotyping using SNPs and/or amino acid polymorphisms at ~20 loci in the SARS-CoV-2 genome was used to differentiate closely-related variants that caused local [10] and global outbreaks [18]. This differentiation obtained by multilocus genotyping was comparable to that produced by phylogenomic analysis. Further refinements in loci selection and integration of population genetic approaches could make multilocus genotyping an attractive genetic tool for viral surveillance. The utility of this tool for surveillance of variant-specific infections associated with different demographics, geographical regions, and different clinical forms of COVID-19 at continental and global scales remains to be demonstrated.

Since the pandemic begun, thousands of SARS-CoV-2 genomes have been made publicly available in the Global Initiative on Sharing Avian Influenza Data (GISAID) database [20]. In this study, we utilised this resource to develop a proof-of-concept multilocus genotyping tool, based on a panel of genome-wide amino acid polymorphisms detected in 22,164 SARS-CoV-2 whole genomes. We demonstrated the utility of this genetic tool in combination with population genetic approaches to monitor viral transmission dynamics and evolution globally at two time points: December 2019 to September 2020 (“2020 genomes”) and January to March 2021 (“2021 genomes”). In doing this, we uncovered contrasting epidemiology and evolution of viral populations within and between continents.

## 2. Material and Methods

### 2.1. Data Curation

All the data used for this study were obtained from GISAID [20]. At the time of our first sampling for this study, September 2020, there were < 20,000 genome sequences of SARS-CoV-2 curated in GISAID. To evaluate the epidemiological diversity and population genetics of SARS-CoV-2 clinical infections, we selected only whole genomes isolated from human infections that were complete and assembled with high-coverage sequence reads. Additionally, these genomes had information on where and when the infection was diagnosed (collection date and country), and the patient’s age and gender. After curation, 5959 whole genomes (referred to as “2020 genomes”) met our criteria and were included in this study (Appendix A). Of these, 66.1%, 48.2%, and 99.9% had information on the clinical outcomes, the specimen type, and the sequencing chemistry used, respectively. An additional 16,205 whole genomes curated in GISAID between 15 January to 15 March 2021 (referred to as “2021 genomes”) which met the inclusion criteria were included. Thus, a total of 22,164 genomes were included in this study.

### 2.2. Study Variables

Individuals from whom the infections were isolated and sequenced were referred to as “patients” for the purpose of this study. Based on the available metadata associated with the 2020 genomes, patient age was stratified into four categories: 0–19, 20–39, 40–59, and ≥ 60 years, as described elsewhere [21]. Clinical cases were grouped as asymptomatic (no symptoms), mild, or severe/critical, as described previously [19]. The specimen type was grouped as upper respiratory tract (URT) or lower respiratory tract (LRT). The country where the infection was diagnosed and/or the genome was isolated was assigned to one of the six continents: Africa, Asia, Europe, Oceania, North America, and South America.

### 2.3. Sequence Alignments and Multilocus Genotyping

All genomes were aligned against the SARS-CoV-2 Wuhan reference strain (NC_045512.2) using minimap version 2.17 [22]. Amino acid changes at codons/loci with SNPs were called using the Geneious Prime SNP caller [23]. Amino acids identical to the reference strain were considered wild-type, else, they were considered mutants. At a codon (locus), an allele (amino acid) was designated by the amino acid letter followed by the codon number. For example, at locus 614 in the S gene, the wild-type allele was indicated as S D_614_ and mutant allele as S G_614_. Contiguous alleles at the loci investigated in a gene were considered as the genotype and those across ≥ 2 genes were considered the multilocus genotype (MLG) for a SARS-CoV-2 genome. Only loci with ≥ 2 alleles, each with a minor allele frequency of 0.01, were used for the MLG construction. These criteria were implemented to ensure unbiased construction of MLGs [24].

### 2.4. Genetic Diversity Indices

Within a continent, the number of unique MLGs, expected number of unique MLGs or eMLGs (i.e., normalised number of MLGs based on smallest sample size), and expected heterozygosity or *H_e_* (estimates genetic diversity, with scores ranging from 0 indicating all MLGs are identical to 1 indicating all MLGs are unique) were estimated using *poppr* V2.8.5 [24]. To assess whether a majority of unique MLGs in a continent were sampled, the eMLG was plotted against the number of genomes sampled as a rarefaction curve using the R package *vegan* [25]. The evenness (E5) statistic was used to evaluate whether the unique MLGs within a continent were evenly distributed. Its score ranges from 0 (presence of predominant MLGs) to 1 (MLGs are evenly distributed).

### 2.5. Linkage Disequilibrium Estimates

To determine which loci in the SARS-CoV-2 genome were co-evolving, i.e., whether infections carried multiple mutations at different loci that were non-randomly associated, we used all the MLGs sampled in a continent to estimate a standardised index of association ($\bar{r}$d) implemented in *poppr*. The $\bar{r}$d is an estimate of linkage disequilibrium (LD), which is the non-random association of alleles at two or more loci. The presence of identical MLGs (i.e., clones) within a population can overestimate the multilocus LD [26]. To account for this, the multilocus LD was clone-corrected ($\bar{r}$d-cc) using only the unique MLGs. To determine which pairs of loci or genes were contributing to the multilocus, i.e., genome-wide LD, a pairwise LD analysis was performed as described elsewhere [27]. The $\bar{r}$d score ranges from 0 to 1 where 0 indicates no LD (i.e., alleles are randomly associated) and 1 indicates complete LD (i.e., alleles are non-randomly associated). The statistical significance of the score was supported by a *p*-value < 0.05.

### 2.6. Genetic Differentiation Estimates

To determine whether SARS-CoV-2 infections from different continents could be genetically differentiated based on the MLGs carried by the SARS-CoV-2 genomes, we estimated the Nei G_ST_ implemented in *mmod* [28]. The G_ST_ score ranges from 0 (no genetic differentiation, i.e., populations are similar or have identical MLGs) to 1 (i.e., complete genetic differentiation, i.e., populations are dissimilar or have unique MLGs). G_ST_ values ranging from 0 to 0.09, 0.1 to 0.19, and ≥0.2 indicate little, moderate, and great genetic differentiation, respectively [29]. We also performed a discriminant analysis of principal components (DAPC), which is a multivariate method for identifying genetic clusters of closely related MLGs [30]. Briefly, the MLG dataset was trained on a K-means algorithm implemented in *adegenet* [30] to identify the optimum number of genetic clusters within the global population. The DAPC was then performed on the genetic clusters retained during a principal component analysis (PCA) by maximizing the genetic variance between populations while minimizing the variance within populations [31]. By adding the information on the continent where each MLG (genome) originated, the DAPC predicted the percentage of relatedness each MLG in a continent had to other MGLs from the other continents, termed “population membership”. This assigned population membership probability was then plotted as a stacked bar using *ggplot2* [32]. This probability, expressed as a percentage, predicted the likelihood of an MLG originating from any of the six continents against the backdrop of the reported continent of origin. To visualise the genetic relationships and clonal complexes among the unique MLGs in the global population, the goeBURST FULL MST algorithm implemented in *Phyloviz* V2 was used to construct networks of minimum spanning trees [33].

### 2.7. Statistical Analysis

Statistical analysis was performed in R v3.5.2 [34] and STATA v16 [35]. Proportions were compared using the chi-square or Fisher’s exact test. Multiple testing was adjusted for using the Holm–Bonferroni method. Logistic regression, using the cluster variance (VCE) method, was performed to determine the risk (odds ratio or OR) of an infected person harbouring SARS-CoV-2 infections that carried mutations at the investigated loci. Age, continent, and gender were considered possible confounders and were adjusted for in the final model. The adjusted OR was considered statistically significant for all analyses where the *p*-value was <0.05.

## 3. Results and Discussion

### 3.1. Demographics of the Study Population

The “2020 genomes” were obtained up to and including September 2020, i.e., pre-vaccine introduction in most countries [15]. At this point in time there were less than 20,000 genomes deposited in GISAID, of which 5959 met our inclusion criteria. The demographics of this study population are described in Table 1. The majority of genomes (>31.0%) were reportedly isolated from SARS-CoV-2 infections in COVID-19 patients in the 40–59 years age group, except in Africa and Oceania where the majority of genomes (≥38.5%) were collected from patients in the 20–39 years age group (Table 1 and Appendix A). In each continent, the majority of genomes were reportedly collected from male patients (≥51.1%), except in Africa where a significantly higher proportion were collected from females (56.2%, *p*-value = 0.002). The majority of SARS-CoV-2 genomes reportedly originated from Asia (26.5%) and <10% originated from Africa, North America, and South America each (Table 1). Overall, ~98% of the genomes were isolated from symptomatic infections including mild and severe COVID-19 cases (Table 1 and Appendix A), demonstrating the early focus on sequencing samples from patients with clinical disease. In most regions except South America, throat swabs constituted > 60% of the clinical samples collected for isolating the virus, and >60% of the viral isolates were sequenced using Illumina. In South America, 60.3% of the isolates were collected from nose swabs and the majority (43.1%) were sequenced using Nanopore (Appendix A).

### 3.2. Polymorphic Loci Investigated

The “2020 genomes” comprised seven GISAID clades: G (22%), GH (23.3%), GR (25.3%), L (4.3%), O (12.1%), S (10%), and V (3%). Seventy-four loci with SNPs that resulted in amino acid changes met the inclusion criteria for genotype and MLG construction. They were in 14 protein coding genes: *orf1ab* polygene (NSP1, NSP2, NSP3, NSP4, NSP5, NSP8, NSP12, NSP13, and NSP14), *orf3a*, *orf8*, M, N, and S genes (Appendix A). Loci including *Orf1ab* 2796 (in NSP3), N 203, S 320, S 477, and *orf3a* 251 were the most polymorphic with five alleles each (Appendix A). Of the 137 mutant alleles detected at the 74 loci, the majority (55%) were in the *orf1ab* polygene, which is ~21 kb (Appendix A), but the average heterozygosity in this gene was lower compared to *orf3a*, S, and N genes (*H_e_* ≤ 0.44 vs. *H_e_* ≥ 0.50). This suggests that the majority of mutations in the *orf1ab* polygene were not under strong selection compared to those in the structural proteins, including the S protein [11,36,37].

## 4. SARS-CoV-2 Transmission Dynamics and Epidemiological Risk of Infection in 2020

The majority of the *orf1ab* mutant alleles were detected infrequently with prevalences below 25% during the early phase of the pandemic, i.e., Feb–May 2020, and thereafter declined sharply as transmission peaked in all the six continents (Appendix A). In contrast, eight mutant alleles were detected at prevalences above 35% and recurred from February to September 2020 in all six continents (Figure 1). They included three in the replicase polygene, *orf1ab*-I_265_ (located in NSP2; an endosome-associated protein that interacts with host proteins), F_3606_ (located in NSP5; encodes the 3CL protease that cleaves the ORF1ab polyprotein), and L_4715_ (located in NSP12; encodes RNA-dependent RNA polymerase, i.e., RdRP, involved in viral replication). The remaining five included two in the accessory proteins, *orf3a* H_57_ and *orf8* S_84_, and three in two structural genes, S G_614_ and N KR_203&204_ (this genotype defines variants in the GR clade). These mutant alleles have been associated with infectivity, disease severity, and immune dysregulation [12,15,17,38] and have been detected in VOCs including the Delta variants [39,40]. In particular, four VOC-defining mutant alleles, *orf1ab* L_4715_, S G_614_, and N KR_203&204_ [16], rose sharply in prevalence from <6% in February 2020 to >90% in August 2020 (Figure 1). This rise coincided with the period when COVID-19 cases, associated with the Delta variant, peaked globally [41], suggesting that these mutant alleles confer a higher transmission advantage than wild-types [19]. Interestingly, ~1% of the Australian genomes sampled in June 2020 carried the S Y_501_ mutant allele, which was associated with high infectivity among UK variants that caused nationwide outbreaks in September 2020 [42]. Our data suggest that the S N501Y mutation emerged earlier, possibly May–June 2020, when Australia had its second wave of transmissions [43].

We investigated the epidemiology of infections with SARS-CoV-2 variants that carried mutant alleles at eight loci, *orf1ab* (265, 3606, and 4715), *orf3a* 57, *orf8* 84, N (203 and 204), and S 614, considered informative for clade/variant designation [16,17] (Figure 2). When compared to younger patients (<20 years), older patients (>20 years) had a higher risk of being infected with variants carrying mutant alleles at four loci, *orf8* 84 (North America), N 203 and 204 (Asian), and *orf3a* 57 (Europe) (Figure 2). Particularly, in Asia and Europe, compared to females, males were more likely to harbour variants with the N KR_203/204_ genotype, associated with Gamma variants/GR clade. Although severe cases compared to asymptomatic cases were more likely to harbour infections that carried mutant alleles at *orf1ab* 265 (in South America), *orf1ab* 3606 (Africa and Asia), *orf3a* 57 (Africa and South America), and *orf8* 84 (Africa and North America), they were less likely to harbour infections that carried mutant alleles at S 614 (Figure 2). Nonsynonymous mutations at the former four loci have been associated with severe disease in previous studies that used a similar dataset from GISAID [17,44], but our data suggested that this association was geography-specific. It is not clear whether this epidemiological risk for infection can be explained by host factors, but host genetics have been associated with COVID-19 severity among different ethnicities [15,44,45]. Our population-level data also support previous in vitro data that demonstrated that the D614G mutation was not associated with severe disease [45]. Different public health interventions may have likely impacted transmission of mutant alleles in different geographical regions [3].

## 5. Population Genetics of SARS-CoV-2 Infections in 2020

We constructed a multilocus genotype (MLG) for each of the 22,164 genomes included in this study to evaluate the utility of the 74 polymorphic loci as a genetic tool for differentiating SARS-CoV-2 variants and for monitoring their evolution in different geographical regions. Among the 5959 genomes in the 2020 study population, 472 unique MLGs were detected in the global population, with MLG repertoire sizes ranging from 40 for South America to 185 for Asia (Appendix A). The mean E.5 score of 0.34 for the global repertoire indicated the presence of predominant MLGs (Table 2). As an example, in each continent, at least the first six most prevalent MLGs were shared by more than 50% of the genomes sampled (Appendix A), suggesting that the majority of clinical cases in 2020 were caused by variants with these and/or closely related MLG signatures.

Among the continents, multilocus genetic diversity in the 2020 study population was highest in Asia and Oceania (eMLGs = ~78, *H_e_* ≥ 0.26) and lowest in South America (eMLGs = 40, *H_e_* = 0.15) (Table 2). This pattern of multilocus diversity was associated with the number of genomes sampled. To assess whether we had captured the majority of the multilocus diversity, we rarefied the number of genomes sampled to the number of unique MLGs detected. The rarefaction curves showed little-to-no sign of plateauing for the number of unique MLGs identified in each continent (Figure 3A). This observation suggested that other unique MLGs existed in the viral population, i.e., in clinical infections, but were not captured in the genomes we sampled in each continent. This underscores the importance of deep sampling during genomic surveillance. It is possible that the undetected MLGs were present in asymptomatic infections, which accounted for ~2% of the genomes that we sampled. Globally, asymptomatic infections constitute ~80% of all COVID-19 cases [8] and as such it is crucial to include them in future genomic surveillance activities.

Significant multilocus LD was detected in the global and continental MLG repertoires ($\bar{r}$d ≥ 0.023, *p*-value < 0.001) (Table 2), which suggested non-random association of alleles among the MLGs at the investigated loci. However, because LD can be overestimated in a population where some genomes carry identical MLGs, we repeated the analysis using only the unique MLG repertoire (Appendix A). Following this, the estimated LD value was attenuated ($\bar{r}$d ≤ 0.014, *p*-value ≥ 0.107) in Africa, Europe, and South America (Table 2), which indicated clonal transmission of SARS-CoV-2 infections in these regions. For example, in Europe and South America, two (EU_2020_MLG1-2) and three (SA_2020_MLG1-3) MLGs, respectively, were predominant and were shared by more than 45% of the genomes sampled (Appendix A). Indeed, multiple outbreaks in Europe and South America were associated with clusters of closely related infections [46,47].

Interestingly, although several of the genomes carried multiple mutations across ≥ 2 genes, pairwise LD estimates showed that the multilocus LD was driven by key mutant alleles that likely co-evolved (Figure 3B and Appendix A). That is, LD was “structured” and existed between specific loci across the genome with respect to the 74 loci. We detected the strongest LD signal ($\bar{r}$d ≥ 0.200, *p*-value < 0.001) between NSP12 and ORF8, NSP12 and S, ORF8 and S, and ORF3a and NSP2 (Figure 3B), which was consistent with previous reports using nucleotide data [48]. Whereas the latter two LD structures either decayed or were maintained depending on the geographical region, the former two were prominent globally (Appendix A). A potential driver of this LD could be the co-selection of the *orf1ab* L_4715_ plus S G_614_ and/or ORF8 S_84_ mutant alleles, particularly in European and North American variants [49]. This explanation was supported by the existence of LD ($\bar{r}$d ≥ 0.10, *p*-value ≤ 0.01) among their loci. Co-selection of mutant alleles across multiple genes may indicate potential recombination signatures and and/or epistatic interaction between specific gene pairs to enhance SARS-CoV-2 fitness [48,49].

## 6. Genetic Clustering and Differentiation of SARS-CoV-2 Populations in 2020 and 2021

To demonstrate the utility of the multilocus genotyping tool for differentiating SARS-CoV-2 variants responsible for COVID-19, we employed a network analysis, DAPC, and G_ST_ to visualise relationships, identify genetic clusters, and to estimate the genetic differentiation among MLG repertoires from the six continents. The network analysis indicated that 11 major clonal complexes (i.e., related MLGs) existed in the “2020 genomes” (Figure 3C). Two of these complexes were unique to Asia (AC1-2) and one was unique to Oceania (OC1); in Australia, a number of SARS-CoV-2 infection clusters were shown to have evolved locally [50]. The remaining eight complexes (GC1-8) consisted of MLGs from the six continents that were indicative of admixture populations (Figure 3D), corroborating previous findings [49]. Among these admixture populations, the DAPC analysis predicted 20–50% relatedness to Asian and Oceanian MLGs (Figure 3D). That is, 20–50% of the alleles in an MLG signature were identical to those in the Asian and Oceanian MLGs. This suggested that the majority of genomes in the global population were related to those from Asia and Oceania. In the early stages of the pandemic, SARS-CoV-2 transmission in Africa was thought to have been seeded by imported cases from America, an assumption that was based on travel data [51,52]. Our DAPC analysis for African MLGs predicted 50–60% relatedness to European MLGs (Figure 3D). As shown in Figure 3D, the majority of the 74 unique African MLGs had 50% of their bars coloured green, representing Europe. In the same plot for Africa, it was observed that <25% of the population membership was assigned to Oceania (indicated in the magenta bar) and Asia (indicated in the blue bar). In summary, these predictions suggest that early transmission in Africa was seeded by infections from multiple geographical origins including Europe. This underscores the need to build strong surveillance systems using both travel and genomic data.

Despite the genetic relatedness among the MLGs in the “2020 genomes”, a number of mutant alleles were unique (i.e., private) to each continent (Appendix A), which indicated the existence of geographical structuring in the 2020 global viral population. Indeed, PCA partitioned all the MLGs in the 2020 study population into four major genetic clusters, indicative of geographical clusters (Figure 3E). One cluster consisted of a subset of MLGs from the six continents, which is consistent with the previously described admixture populations in Figure 3D. In contrast, the other three clusters depicted continental clusters that were made up of MLGs, predominantly from Africa, Asia, and North America (Figure 3E). A minor proportion of MLGs from Oceania and Europe showed clinal differentiation into North America and Africa, respectively (Figure 3E), which likely represent closely related SARS-CoV-2 variants that spread between these continents [53]. Geographical clustering in the “2020 genomes” was further supported by the G_ST_ estimates, which indicated moderate-to-high genetic differentiation (G_ST_ ≥ 0.204) except between Oceania and Asia, Oceania and Europe, and Europe and South America (Figure 3F). In addition, we observed moderate-to-high genetic differentiation (G_ST_ ≥ 0.111) within continents (Appendix A), demonstrating the potential utility of the 74 loci for surveillance at the regional level.

We then tested the versatility of our multilocus genotyping tool to differentiate SARS-CoV-2 genomes with respect to their GISAID clade designation [6]. The 16,205 genomes sampled for the “2021 genomes” (i.e., 2021 study population) comprised nine GISAID clades: G (8.3%), GH (24%), GK (0.1%), GR (29.7%), GRY (29.2%), GV (6.3%), L (0.2%), O (0.6%), and S (1.6%). For this population with a total of 445 unique MLGs, the PCA showed considerable genetic differentiation among the clades (Figure 4A). During the first quarter of 2020 at the start of the pandemic, Tang and colleagues proposed the S and L clade nomenclature for differentiating SARS-CoV-2 infections [54], with the L clade later splitting into G and V as novel variants began to emerge [6]. Four clades that arose from the G clade (characterised by the D614G mutation) including GH (ORF3a Q57H), GK (S T478K), GR (N G203R), and GV (S A222V) formed four separate clusters in the PCA for the 2021 study population (Figure 4A). The GRY clade, which was absent in the “2020 genomes”, showed clinal differentiation into the GR clade, with the former splitting from the later around September 2020 by acquiring deletions such as S (H69, V70 and Y144) and substitutions including S N501Y and N G204R [55]. VOCs including the Alpha, Beta, Gamma, and Delta variants, associated with the GRY, GH, GR, and G/GK clades, respectively, were responsible for large waves of SARS-CoV-2 transmission and COVID-19 cases in most parts of the world in 2021 [56,57].

Furthermore, the PCA also separated the MLGs in the “2021 genomes” based on the continent of origin (Figure 4B). The majority of MLGs from South America and a subset of those from North America formed two well-defined clusters whereas European and Asian MLGs clustered with the African MLGs. This pattern of clustering contrasts the pattern observed for the 2020 study population, suggesting that the two viral populations were genetically distinct (Appendix A). Indeed, the DAPC analysis indicated the existence of predominant MLGs in the 2020 global population including Africa (number of predominant MLGs = 2; detected in 27.6% of the African genomes sampled), Asia (4; 36.4%), Europe (2; 45.9%), North America (3; 55.1%), and South America (3; 54.4%). These viral populations were genetically differentiated (G_ST_ ≥ 0.333) from their respective 2021 viral populations: Africa (3; 66.2%), Asia (2; 54.3%), Europe (2; 70.0%), North America (3; 76.3%), and South America (2; 83.3%) (Figure 4C–G and Appendix A). Oceania had three predominant MLGs that accounted for 34.4% of the genomes sampled in 2020 but we could not compare these to the 2021 viral population since no genomes were available between 15 January and 15 March at the time of sampling. Taken together, our findings from the 2020 and 2021 viral genomes suggest that in the different continents, COVID-19 cases in 2020 and 2021 were caused by genetically distinct viral strains that likely adapted to local populations [58].

## 7. Conclusions

Genomic tools are needed to differentiate SARS-CoV-2 variants causing COVID-19 and to monitor their evolution in different geographical regions. Here, we identified 74 polymorphic loci located in 14 genes in 22,164 SARS-CoV-2 whole genomes isolated from clinical infections that occurred in 2020 and 2021. The low prevalence of mutant alleles in the NSPs and the low genetic diversity in the *orf1ab* polygene in comparison to the structural genes including N and S suggested that the majority of *orf1ab* mutations were not under strong selection in the 2020 viral population. In contrast, the predominance and spatiotemporal persistence of the eight mutant alleles, N KR_203&204_, *orf1ab* (I_265_, F_3606_ and L_4715_), *orf3a* H_57_, *orf8* S_84,_ and S G_614,_ indicate their selective advantage. As demonstrated by the detection of significant multilocus LD, SARS-CoV-2 variants including the VOCs may have acquired multiple co-evolving mutations to enhance their transmissibility. Epidemiologically, the risk for variant-specific infections among vulnerable populations may vary across different demographics and geographical regions, intimating that future control interventions need to target all age groups.

Although extensive sampling of symptomatic infections is encouraged during surveillance activities, it is paramount that asymptomatic infections, which constitute the majority of COVID-19 cases, are included during surveillance in order to detect novel variants. Among the 2020 and 2021 variants that caused COVID-19, our multilocus genotyping tool identified continental clusters, which suggested geographical structure in the global viral population. Furthermore, the DAPC results suggested that the majority of infections in 2021 and 2021 were caused by genetically distinct variants. In particular, in the 2021 viral population, we observed considerable genetic differentiation among variants in the GISAID clades including GRY (Alpha), GH (Beta), GR (Gamma), and G/GK (Delta variant), which were largely responsible for successive waves of transmission and COVID-19 outbreaks in multiple countries. These findings are consistent with previous reports using phylogenomic analysis, and thus demonstrate the utility of the 74 polymorphic loci as a proof-of-concept multilocus genotype tool for monitoring the viral evolution. However, the limited genetic differentiation of the admixture populations suggest that more polymorphic loci may be needed to accurately differentiate closely related variants.

## 8. Limitations

The samples used in this study reflect both health policy decision-making and sampling strategies implemented in the countries at the time. The data presented in this study, though they are based on publicly available data and are thus relevant, need further investigations to draw definite conclusions on the associations between age, gender, and geographical region and the SARS-CoV-2 variant causing COVID-19. Nearly all the genomes sampled lacked metadata on country of infection, instead showing the country of detection, making it difficult to infer a source of infection based solely on the MLG data. Furthermore, the LD structures detected in this study population may change as the virus continues to evolve, particularly in response to control interventions such as vaccines. Thus, continuous genomic surveillance is warranted. We look forward to future studies aiming to improve the MLG genotyping tool. Such studies would need to factor in polymorphisms in the most recently emerged variants, including SARS-CoV-2 Omicron variants. This study did not include the Omicron genomes since, at the time of the study from January 2020 to March 2021, they were not publicly available. The Omicron variants emerged in November 2021.

## Figures and Tables

**Figure 1 viruses-14-01434-f001:**
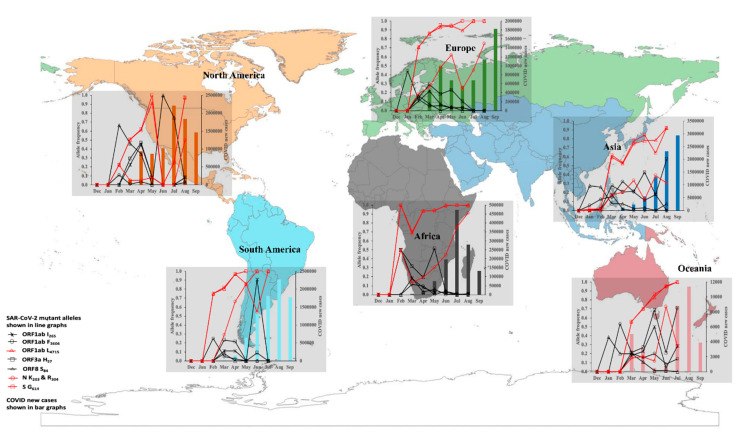
Longitudinal prevalence of eight SARS-CoV-2 mutant alleles and reported new cases of COVID-19 in 2020. The prevalence data for the alleles and newly confirmed cases (WHO report 2020) are reported for December 2019 to September 2020. The frequency of the eight mutant alleles is shown with the line graph. Four mutant alleles, *orf1ab* L_4715_, S G_614_, and N K_203,_ and R_204_, were associated with sharp peaks in the number of COVID-19 new cases (bar graph).

**Figure 2 viruses-14-01434-f002:**
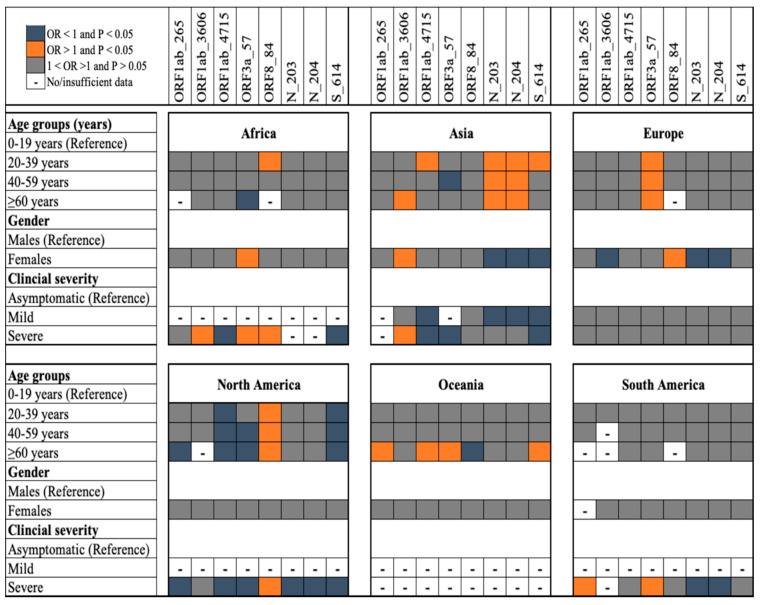
Epidemiology of infection with variant-specific alleles at eight informative loci in SARS-CoV-2 genomes sampled in 2020. The likelihood of patients harbouring an infection that carried a mutant allele at the eight informative loci was compared among patients in different age groups, gender, and geographical regions and patients with different COVID-19 forms. The adjusted odds ratio (OR) with the *p*-value is depicted with orange, blue, and grey colours corresponding to OR > 1 and *p*-value < 0.05; OR < 1 and *p*-value < 0.05; and 1< OR > 1 and *p*-value > 0.05, respectively. – indicates no data, i.e., the OR was not estimated due to the sample size being less than five.

**Figure 3 viruses-14-01434-f003:**
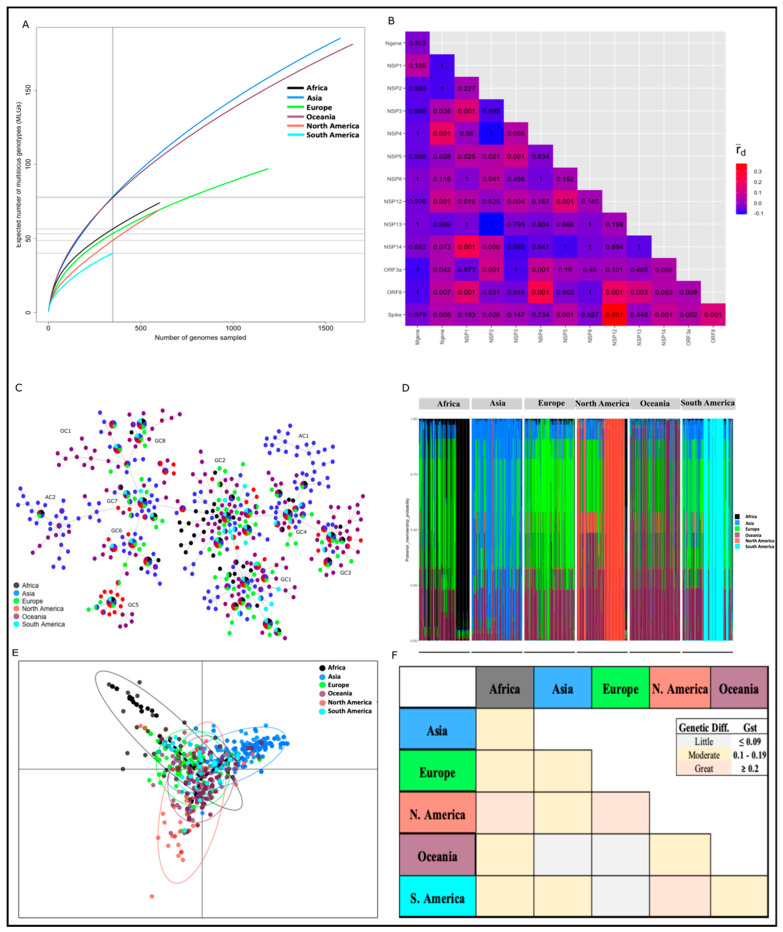
Multilocus LD, genetic relatedness, and differentiation of MLGs in the 2020 study population. (**A**). Rarefaction curve of the unique MLGs identified in each geographical region. There was no sign of levelling-off, i.e., plateauing in the curves for all six continents, indicating that not all unique MLGs in each continent were detected. (**B**). Pairwise LD estimates among genes in the 2020 study population. The $\bar{r}$d ranges from 0 (no LD) to 1 (complete LD). The values in the coloured heatmap indicate the *p*-value associated with the pairwise $\bar{r}$d estimates. The strongest LD signal ($\bar{r}$d ≥ 0.200, *p*-value < 0.001) was observed between NSP12 and ORF8, NSP12 and S, ORF8 and S, and ORF3a and NSP2. (**C**). Network analysis to visualise the relatedness among the 472 unique MLGs detected in the 2020 study population. The minimum spanning tree identified 11 clonal complexes including eight global complexes (GC1–8) and three continent-specific complexes, Asia (AC1-2) and Oceania (OC1). (**D**). Population membership probability assignments of MLGs in each continent. This probability, expressed as a percentage, predicted how closely related MLGs were to each other with respect to the reported continent of origin. Admixture populations were prominent in all six continents. (**E**). DAPC analysis identified one global cluster (MLGs from all continents, central axis of PCA plot) and three continental clusters, Africa, Asia, and North America. (**F**). Genetic differentiation (Nei G_ST_) of MLGs in the 2020 study population. G_ST_ values ranging from 0 to 0.09, 0.1 to 0.19, and ≥0.2 indicate little, moderate, and great genetic differentiation, respectively.

**Figure 4 viruses-14-01434-f004:**
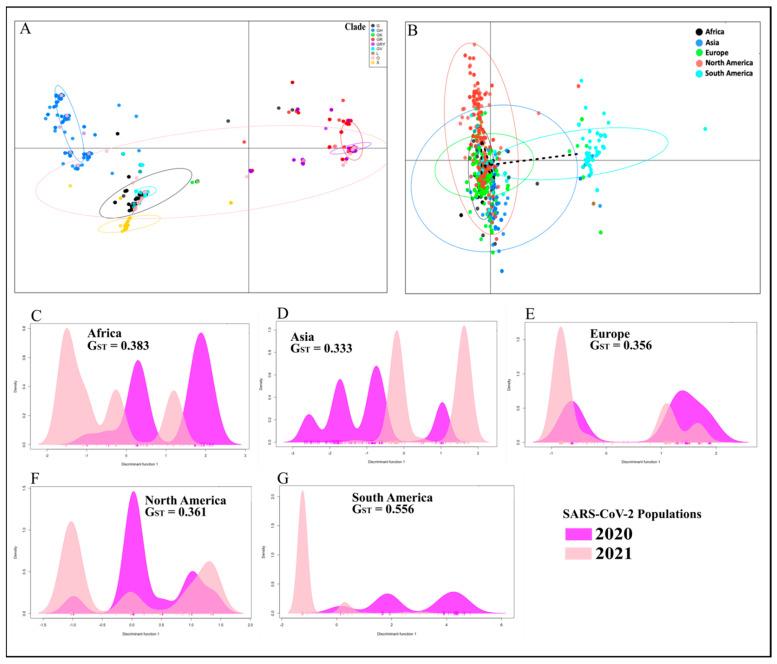
Multilocus genetic differentiation of 2020 and 2021 MLG repertoires. For the 2020 and 2021 viral populations that were sampled, 472 and 445 unique MLGs, respectively, were detected. (**A**). Among the 2021 MLGs, the PCA showed considerable genetic differentiation with respect to the nine GISAID clades: G, GH, GK, GR, GRY, GV, L, O, and S. (**B**). Clades GH (ORF3a Q57H), GK (S T478K), GR (N G203R) and GV (S A222V), which arose from the G clade, formed separate clusters. VOCs including the Alpha, Beta, Gamma, and Delta variants, associated with the GRY, GH, GR, and G/GK clades, respectively, were responsible for large waves of SARS-CoV-2 transmission and COVID-19 cases in most parts of the world in 2021. (**C**–**G**). Predominant MLGs (represented by the coloured peaks) accounted for the majority of genomes sampled in the 2020 and 2021 viral populations; within each continent, there was great genetic differentiation (G_ST_ ≥ 0.333) between the 2020 and 2021 MLG repertoires.

**Table 1 viruses-14-01434-t001:** Demographics of the 2020 study population.

		Global	Africa	Asia	Europe	North America	Oceania	South America
Characteristics		5959	601	1579	1188	597	1646	348
**Age group ^#^ (years)**	0–19	323 (5.4)	68 (11.3)	123 (7.8)	43 (3.6)	20 (3.6)	58 (3.5)	11 (3.2)
	20–39	2054 (34.5)	276 (45.9)	559 (35.4)	273 (22.9)	196 (32.8)	634 (38.5)	116 (33.3)
	40–59	1996 (33.5)	185 (30.8)	561 (35.5)	389 (32.7)	217 (36.4)	515 (31.3)	129 (37.1)
	60+	1586 (26.6)	72 (11.9)	336 (21.3)	483 (40.7)	164 (27.5)	439 (26.7)	92 (26.4)
**Gender ^#^**	Females	2664 (44.7)	338 (56.2)	581 (36.8)	581 (48.9)	250 (41.9)	754 (45.8)	160 (45.9)
	Males	3295 (55.3)	263 (43.8)	998 (63.2)	607 (51.1)	347 (58.1)	892 (54.2)	188 (54.0)
**Clinical ^#^ Severity**	Asymptomatic	60 (1.5)	0 (0.0)	41 (2.7)	18 (1.9)	0 (0.0)	0 (0.0)	1 (0.4)
	Mild	557 (14.1)	15 (2.5)	134 (8.8)	64 (6.7)	282 (47.2)	2 (12.5)	60 (24.9)
	Severe	3321 (84.33)	586 (97.5)	1356(88.6)	870 (91.4)	315 (52.8)	14 (87.5)	180 (74.7)
	Missing data *	2021	0	48	236	0	1630	107

^#^ Denotes number of genomes and the percentage, N (%). * Denotes the number of genomes with missing clinical data and this was not included in the percentage calculations.

**Table 2 viruses-14-01434-t002:** Genetic diversity estimates for the 2020 SARS-CoV-2 study population.

Continent	N	MLGs	eMLGs (SE)	E.5	*H_e_*	$\bar{r}$d (*p*-Value)	$\bar{r}$d-cc (*p*-Value)
**Africa**	601	74	56.4 (3.1)	0.57	0.19	0.037 (0.001)	0.009 (0.180)
**Asia**	1579	185	78.1(5.2)	0.48	0.26	0.084 (0.001)	0.024 (0.001)
**Europe**	1188	97	53.2 (3.8)	0.41	0.18	0.088 (0.001)	0.011 (0.107)
**North America**	597	69	48.6 (3.3)	0.47	0.30	0.234 (0.001)	0.073 (0.001)
**Oceania**	1646	181	77.5 (5.0)	0.45	0.30	0.087 (0.001)	0.023 (0.001)
**South America**	348	40	40.0 (0.0)	0.60	0.15	0.091 (0.001)	0.014 (0.132)
**Total**	5959	472	95.3 (5.8)	0.34	0.26	0.086 (0.001)	0.023 (0.001)

The number of observed MLGs was normalised by the smallest sample size to obtain the expected MLGs (eMLGs) with the standard error (SE). The standardised index of association ($\bar{r}$d) was clone-corrected ($\bar{r}$d-cc) using the unique MLG dataset.

## Data Availability

See Appendix A.

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
