# Peer review of "Contrasting Epidemiology and Population Genetics of COVID-19 Infections Defined by Multilocus Genotypes in SARS-CoV-2 Genomes Sampled Globally"

_viruses, 2022, doi:10.3390/v14071434_

Round 1

Reviewer 1 Report

The study by Chan et al is interesting and in line with current pandemic challenges. The authors have analyzed the SARS-CoV-2 polpulation genetics in different regions across the globe. They have identified 74 polymorphic loci, located in 14 genes from 22,164 whole genome isolated from clinical infections that occurred from 2020-2021. The works highlights the importance of genomic surveillance os SARS CoV-2 variants. However the authors need to address following comments:

1. The authors have mentioned - dataset analysed in the study was from two time points (Dec19- Sep20) & (Jan-Mar21)- however the study misses on including the omicron variant, as it has been reported to be highly mutable and evade immune response. Hopefully it can further strengthen the significance of the study esp in VOC- defining mutant allele.

2.  Authors need to further elaborate the genetic characterization and differentiation of SARS-CoV-2 esp the DAPC analysis showing relatedness of MLGs? It’s not clear how did they determined the relatedness between different regions like asia & europe or europe & africa.

3. Authors need to elaborate the point mentioned in results section - VOC’s defining mutant alleles those showing significant rise from <6% to ~90% by aug 2020 had selective advantage ?

4. Authors also need to further elaborate the SARS-CoV-2 VOC’s including the omicron variant within the introduction section.

5. Have authors submitted the final data (result, analysis, code, etc) to any of the data repository?  

Author Response

  1. The authors have mentioned - dataset analysed in the study was from two time points (Dec19- Sep20) & (Jan-Mar21)- however the study misses on including the omicron variant, as it has been reported to be highly mutable and evade immune response. Hopefully it can further strengthen the significance of the study esp in VOC- defining mutant allele.

Response: We thank the reviewer for the comment on the omicron variant. The Omicron variants of SARS-CoV-2 will be an exciting group to include in future studies. The window of time we have covered in our research already provides us with several insights into the potential use of the strategies we propose to employ and for the assessment of their insightfulness. We have then decided to include the following statement in the Limitations section to address the reviewer’s comment: “We look forward to future studies aiming to improve the MLG genotyping tool to factor in polymorphisms in the most recently emerged variants, including SARS-CoV-2 omicron variants. This study did not include the omicron genomes since, at the time of the study, from January 2020 to March 2021, they were not publicly available. The omicron variants emerged in November 2021.”

  1. Authors need to further elaborate the genetic characterization and differentiation of SARS-CoV-2 esp the DAPC analysis showing relatedness of MLGs? It’s not clear how did they determined the relatedness between different regions like asia & europe or europe & africa.

Response: Thank you for the opportunity to expand on this topic. The MLG data (74 loci) contains information on the reported continent of origin of the genomes. The DAPC analysis, described in the Methods section and detailed in the cited paper Jombart et al., uses the MLG signatures to determine the percentage of related among the MLGs. By adding the information on the continent where each MLG (genome) originated, the DAPC predicted the percentage of relatedness that each MLG in a continent had to other MGLs from different continents – termed ‘population membership’. These assigned population membership probabilities were then plotted as a stacked bar using ggplot2 [30]. This probability, expressed as a percentage, predicted the likelihood of an MLG originating from any of the six continents against the backdrop of the reported continent of origin. As shown in Figure 3D, most of the 74 unique African MLGs have 50% of their bars coloured green, representing Europe. In the same plot for Africa, it can also be seen that < 25% of population membership is assigned to Oceania (indicated in the magenta bar) and Asia (indicated in blue bar). In summary, these predictions suggest that early transmission in Africa was seeded by infections from multiple geographical origins, including Europe. We thank the reviewer for highlighting this lack of clarity, and we suggest making the following revisions to the manuscript text:

Revised additions

Line 176-181 “By adding the information on the continent where each MLG (genome) originated, the DAPC predicted the percentage of relatedness that each MLG in a continent had to the MGLs from different continents – this is termed ‘population membership’. These assigned population membership probabilities were then plotted as a stacked bar using ggplot2 [30]. This probability, expressed as a percentage, predicted the likelihood of an MLG originating from any of the six continents against the backdrop of the reported continent of origin.”

Line 339-342 “As shown in Figure 3D, the majority of the 74 unique African MLGs have 50% of their bars coloured in green (representing Europe). In the same plot for Africa, it can also be seen that < 25% of population membership is assigned to Oceania (represented in the magenta bar) and Asia (represented by the blue bar).”

DAPC original paper. Jombart, T., S. Devillard, and F. Balloux, Discriminant analysis of principal components: a new method for the analysis of genetically structured populations. BMC genetics, 2010. 11(1): p. 94)

  1. Authors need to elaborate the point mentioned in results section - VOC’s defining mutant alleles those showing significant rise from <6% to ~90% by aug 2020 had selective advantage?

Response: Thank you for this. We have elaborated the findings as requested

Revised addition. Line 246-248. “This rise coincided with the period when COVID-19 cases, associated with the Delta variant, peaked globally [39], suggesting that these mutant alleles confer a higher transmission advantage than wild-types [17]”.

  1. Authors also need to further elaborate the SARS-CoV-2 VOC’s including the omicron variant within the introduction section.

Response: We have clarified the original statement and included omicron in the VOCs.

Revised addition.

Line 57-58. “i.e., variants of concern (VOCs) including Alpha, Beta, Gamma, Delta and Omicron.”

  1. Have authors submitted the final data (result, analysis, code, etc) to any of the data repository?  

Response: All databases and code used for analysis have been included in the manuscript as supplementary data and are thus feely accessible to the audience of the journal.

Reviewer 2 Report

The manuscript describes the utility of multilocus genotyping tool to study amino acid polymorphism in SARS-CoV-2 globally based on analysis of genomes deposited in GISAID database at two time points-2020 and 2021. Results from the study implicates in understanding the viral epidemiology, transmission dynamics and evolution in different populations and regions worldwide. Though the article provides very useful information across continents identifying two clusters.

I think findings can be strengthened by addressing the following comments

1. Genome analysis of sequences deposited from breakthrough infections, especially from the sequences of 2021 genomes. That helps to dissect the amino acid polymorphism in those populations as well as help in identifying new clusters and the efficacy of vaccines.

2. Genome analysis of sequences deposited from people with comorbidities. This helps if new variants' arrival was related to the altered immune status of the host.

3. In the discussion, the authors may explain how variants noted in this study were correlated with increased transmissibility or severity of the infection, in addition to explaining the utility of MLG.

Author Response

The manuscript describes the utility of multilocus genotyping tool to study amino acid polymorphism in SARS-CoV-2 globally based on analysis of genomes deposited in GISAID database at two time points-2020 and 2021. Results from the study implicates in understanding the viral epidemiology, transmission dynamics and evolution in different populations and regions worldwide. Though the article provides beneficial information across continents identifying two clusters.

I think findings can be strengthened by addressing the following comments

  1. Genome analysis of sequences deposited from breakthrough infections, especially from the sequences of 2021 genomes. That helps to dissect the amino acid polymorphism in those populations as well as help in identifying new clusters and the efficacy of vaccines.

Response: The reviewer is very right and touches on a point of great interest. Conducting a sub-group analysis of breakthrough infections could provide crucial data on vaccine efficacy. Unfortunately, the genomes we included in this study did not have any associated data that would have allowed us to conduct such an analysis. This topic, including vaccine resistance is an area that our group will explore in future studies.

  1. Genome analysis of sequences deposited from people with comorbidities. This helps if new variants' arrival was related to the altered immune status of the host.

Response: Once again, we thank the reviewer for this comment. Another observation that we would have liked to be able to make. Unfortunately, and once again, the sub-group analysis by comorbidities was not included in this study simply because none of the genomes included in this study possessed the appropriate metadata. We believe this emphasises the need to collect not only more data but also better data and to make it freely available so that more in-depth studies can be conducted. However, and for the purposes of the aims of our study, the genomes we were able to obtain were still able to validate our approach.

  1. In the discussion, the authors may explain how variants noted in this study were correlated with increased transmissibility or severity of the infection, in addition to explaining the utility of MLG.

Response: We thank the reviewer for this comment and agree that a discussion of the relationship between increased transmissibility or severity of infection with each of the variants is indispensable. We have strived to address that in the original manuscript under the section ‘SARS-CoV-2 transmission dynamics and epidemiological risk of infection in 2020’ which can be found between lines 256-276.

Please note. We have also made minor revisions in the methods section. We hope this makes it clearer.